# Underlying mechanisms of oxygen uptake kinetics in chronic post-stroke individuals: A correlational, cross-sectional pilot study

**Jean Alex Matos Ribeiro**[1], **Acson Gustavo da Silva Oliveira**[1], **Luciana Di Thommazo-Luporini**[1], **Clara Italiano Monteiro**[1], **Gabriela Nagai Ocamoto**[1], **Aparecida Maria Catai**[1], **Audrey Borghi-Silva**[1], **Shane A. Phillips**[2], **Thiago Luiz Russo**[1] *

**1** Department of Physical Therapy, Federal University of São Carlos, São Carlos, Brazil, **2** Department of Physical Therapy, College of Applied Health Sciences, University of Illinois at Chicago, Chicago, Illinois, United States of America

* russo@ufscar.br

**Data Availability Statement:** All relevant data are within the manuscript and its Supporting Information files.

## Abstract

Post-stroke individuals presented deleterious changes in skeletal muscle and in the cardiovascular system, which are related to reduced oxygen uptake ($\dot{V}O_2$) and take longer to produce energy from oxygen-dependent sources at the onset of exercise (mean response time, MTR$_{ON}$) and during post-exercise recovery (MRT$_{OFF}$). However, to the best of our knowledge, no previous study has investigated the potential mechanisms related to $\dot{V}O_2$ kinetics response (MRT$_{ON}$ and MRT$_{OFF}$) in post-stroke populations. The main objective of this study was to determine whether the MTR$_{ON}$ and MRT$_{OFF}$ are related to: 1) body composition; 2) arterial compliance; 3) endothelial function; and 4) hematological and inflammatory profiles in chronic post-stroke individuals. Data on oxygen uptake ($\dot{V}O_2$) were collected using a portable metabolic system (Oxycon Mobile®) during the six-minute walk test (6MWT). The time to achieve 63% of $\dot{V}O_2$ during a steady state (MTR$_{ON}$) and recovery (MRT$_{OFF}$) were analyzed by the monoexponential model and corrected by a work rate (wMRT$_{ON}$ and wMRT$_{OFF}$) during 6MWT. Correlation analyses were made using Spearman's rank correlation coefficient ($r_s$) and the bias-corrected and accelerated bootstrap method was used to estimate the 95% confidence intervals. Twenty-four post-stroke participants who were physically inactive took part in the study. The wMRT$_{OFF}$ was correlated with the following: skeletal muscle mass ($r_s$ = -0.46), skeletal muscle mass index ($r_s$ = -0.45), augmentation index ($r_s$ = 0.44), augmentation index normalized to a heart rate of 75 bpm ($r_s$ = 0.64), reflection magnitude ($r_s$ = 0.43), erythrocyte ($r_s$ = -0.61), hemoglobin ($r_s$ = -0.54), hematocrit ($r_s$ = -0.52) and high-sensitivity C-reactive protein ($r_s$ = 0.58), all p < 0.05. A greater amount of oxygen uptake during post-walking recovery is partially related to lower skeletal muscle mass, greater arterial stiffness, reduced number of erythrocytes and higher systemic inflammation in post-stroke individuals.

**Funding:** This study was funded by the Brazilian Government Funding Agencies: Coordination for the Improvement of Higher Education Personnel – CAPES (Finance Code 001), the São Paulo Research Foundation – FAPESP (funding: 2017/13655-6 and 2017/22173-5) and the National Council for Scientific and Technological Development – CNPq (funding: 442972/2014-8).

**Competing interests:** The authors have declared that no competing interests exist.

## Introduction

Standing up and walking to the workplace requires proper oxygen uptake ($\dot{V}O_2$), which is regulated over time by well-controlled mechanisms. Thus, $\dot{V}O_2$ kinetics reflects the efficiency of pulmonary, cardiovascular and skeletal muscle systems' interaction during physical activity [1]. Sustained submaximal physical activities, such as walking, require a steady-state $\dot{V}O_2$ ($\dot{V}O_{2SS}$). The time interval between oxygen at rest ($\dot{V}O_{2REST}$) and the $\dot{V}O_{2SS}$ is usually expressed by the mean response time ($MRT_{ON}$) that represents the body's ability to uptake the oxygen quickly enough in order to produce energy for movement [1]. On the other hand, whether the ending of the activity is considered, the time interval between $\dot{V}O_{2SS}$ and $\dot{V}O_2$ during post-activity recovery ($\dot{V}O_{2RECOVERY}$) is expressed by $MRT_{OFF}$ that represents the amount of $\dot{V}O_2$ needed to restore the body to its resting level of metabolic function (see Fig 1) [1–3].

A decrease in $MRT_{ON}$ is related to the early use of oxygen-dependent energy sources and is therefore much more energy efficient than oxygen-independent energy sources [1, 4]. Likewise, $MRT_{OFF}$ is an outcome of $\dot{V}O_2$ kinetics to understand the recovery phase when the $\dot{V}O_2$ is used to produce energy related to thermal, hormonal, and metabolic processes, as well as to resynthesize stored creatine phosphate in the muscle and refill oxygen stores in blood and muscle, used during walking [1–3]. Slower $MRT_{ON}$ and $MRT_{OFF}$ are involved with a marked exercise intolerance [1–4]. Thus, understanding the mechanisms limiting $\dot{V}O_2$ is essential for improving bioenergetics kinetics (i.e. $\dot{V}O_2$ on- and off-kinetics), and therefore aerobic endurance [4].

Previous studies have shown that, in post-stroke individuals, both $MRT_{ON}$ and $MRT_{OFF}$ are slower than their healthy matched peers [5, 6], which implies an inefficiency energy production at the onset of exercise and during recovery. In addition, bioenergetic kinetics has been described as a limiting factor in the ability of post-stroke individuals to walk in a real-world environment. Previous studies [5–7] suggest that after chronic stroke, individuals have a sluggish capacity to transport, extract and/or consume oxygen, at the onset of exercise (slow $\dot{V}O_2$ on-kinetics) or during the recovery phase (slow $\dot{V}O_2$ off-kinetics), and this is associated with fewer steps/day and the inability to sustain longer periods of activities in the real world [6, 7].

After a stroke, these individuals have deleterious stroke-related skeletal muscle changes, such as a shift from type I to type II fibers, muscle atrophy, intramuscular fat, and muscle fibrosis [8–11]. In addition, they have stroke-related cardiovascular changes, such as endothelial dysfunction, impaired arterial compliance, and increased proinflammatory markers, which reduce the vasodilators (e.g. nitric oxide) and decrease vessel diameter, consequently, affecting blood flow [9]. In particular, the C-reactive protein level, a marker of systemic inflammation, remains elevated during the chronic phase of stroke [12] and is related to anemia in individuals with chronic inflammatory [13, 14]. These alterations are related to a reduction in $\dot{V}O_2$ [9, 15–17] and are potential targets to understand why bioenergetics kinetics is altered in post-stroke individuals. However, to the best of our knowledge, no study investigated which mechanisms are related to $\dot{V}O_2$ kinetics response in chronic post-stroke population. Thus, the main objective of this study was to determine whether the MRT (on and off) is correlated with: 1) body composition; 2) arterial compliance; 3) endothelial function; and 4) hematological and inflammatory profiles in post-stroke individuals. We hypothesize that the underlying mechanisms mentioned above might be involved with $\dot{V}O_2$ kinetics.

## Methods

### Study design and ethical aspects

This is a correlational, cross-sectional pilot study with a convenience sample (there was no random selection). We followed the STrengthening the Reporting of OBservational studies in

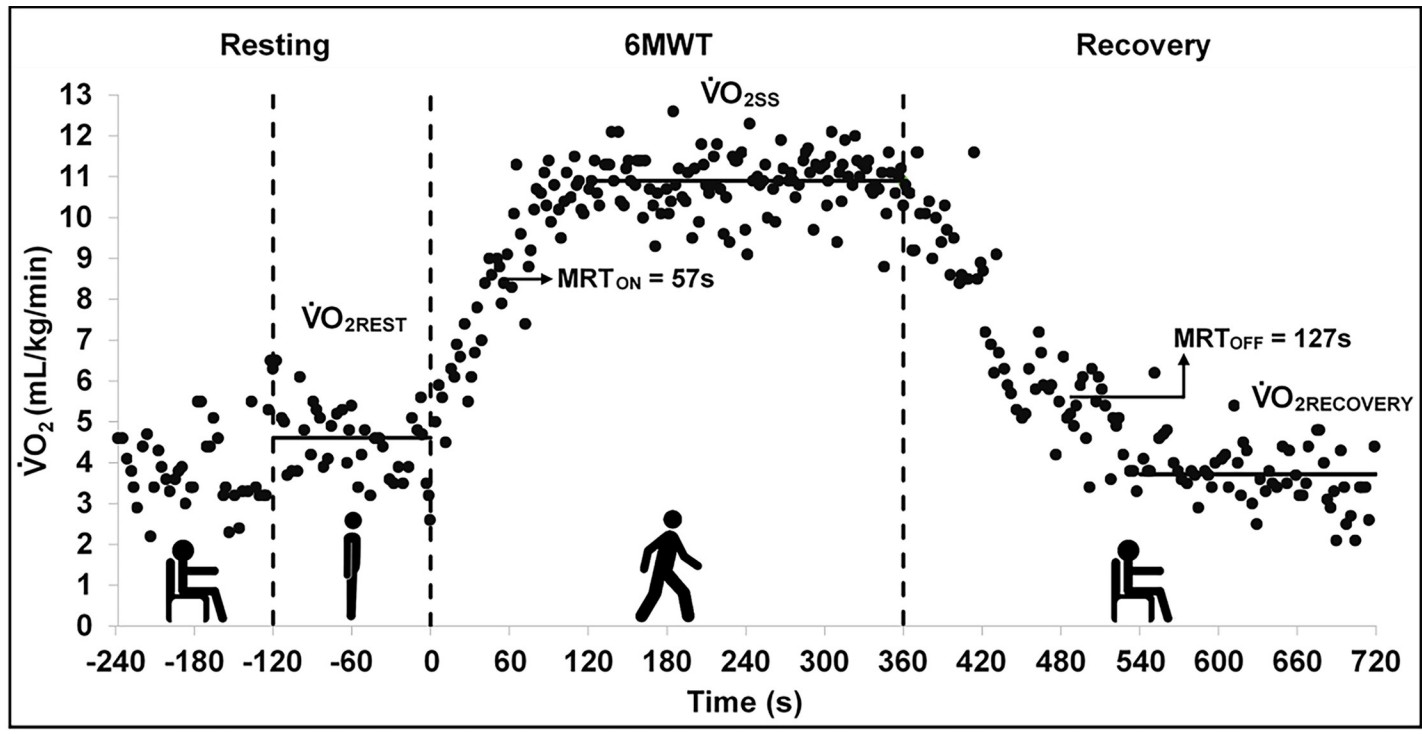

**Fig 1. Oxygen uptake response to the 6-minute walk test.** Oxygen uptake raw data measured breath-by-breath from a sixty-two-year-old woman with stroke and severe motor function impairment. The vertical three dashed lines indicate the sit-to-stand, stand-to-test, and test-to-sit phases, in sequence. Each data point indicates breath-by-breath values averaged every 3 seconds. 6MWT, six-minute walk test; mL/kg/min, milliliter per kilogram per minute; $MRT_{OFF}$, mean response time of oxygen uptake off-kinetics; $MRT_{ON}$, mean response time of oxygen uptake on-kinetics; s, second; $\dot{V}O_2$, oxygen uptake; $\dot{V}O_{2RECOVERY}$, oxygen uptake during the recovery period; $\dot{V}O_{2REST}$, oxygen uptake at rest; $\dot{V}O_{2SS}$, oxygen uptake during effort at steady-state level.

Epidemiology (STROBE) guidelines to report our study methods and results. The study protocol was approved by the Ethics and Research Committee at the Federal University of São Carlos, Brazil [number: 62417216.9.0000.5504]. The funders played no role in the design, conduct, or reporting of this study. All participants gave written informed consent before participating in the study.

## Setting and participants

The participants were recruited between January 2017 and July 2019 from the local community and nearby cities in the state of São Paulo, Brazil. Individuals included were: 1) 40–80 years of age; 2) stroke diagnosis (ischemic or hemorrhagic) confirmed by computer tomography or magnetic resonance imaging; 3) chronic stroke (time since stroke ≥ 6 months); 4) able to walk independently, including those with a need for aids or orthoses (Functional Ambulation Classification ≥ 3) [18]; 5) physically inactive or insufficiently active [International Physical Activity Questionnaire (IPAQ); < 150 min of moderate-to-vigorous-intensity physical activity per week or < 75 minutes of vigorous-intensity physical activity per week or an equivalent combination of moderate- and vigorous-intensity activity] [19, 20]; and 6) absence of cognitive impairment [Mini-Mental State Examination (MMSE); illiterate (≥ 13 points), elementary and middle (≥ 18 points), and high (≥ 26 points) level literacy] [21]. Individuals excluded were: 1) cardiac surgery and or myocardial infarction; 2) uncontrolled chronic disease; and 3) active/passive smoker and/or regular consumer of alcoholic beverages.

All procedures were carried out over three non-consecutive days with a minimum of 72h interval. On the first day of the assessment, participants were interviewed in order to obtain data on personal characteristics, then height and motor function were measured, and the six-minute walk test was performed between 2 and 6 pm at the Department of Physical Therapy at the Federal University of São Carlos, Brazil. A wearable activity monitor was then placed on the participants' nonparetic ankle. On the second day of the assessment, participants returned to the department, the activity monitor was removed, and the body composition, arterial compliance and endothelial function were assessed between 8 and 10 am. On the third day of the assessment, a blood sample was collected from participants between 8 and 10 am at the Clinical Analysis Laboratory at UNIMED (a cooperative medical system) in São Carlos, Brazil.

## Day 1 assessment

**Clinical assessment.** The anthropometric data of height was measured with a standard stadiometer (Welmy R-110, Santa Barbara do Oeste, SP, Brazil). The motor impairment characteristics were measured by the Fugl-Meyer Assessment of Motor Recovery after Stroke (FMA). The motor function domains of the FMA score range from 0 to a maximum of 100 points, and according to their points, participants' motor function was classified as severe ($<50$), marked (50–84), moderate (85–94) or slight (95–99) [22]. A single physiotherapist with a background in FMA conducted all the clinical assessments.

**The Six-Minute Walk Test (6MWT).** The 6MWT was performed according to the American Thoracic Society standards [23], except that individuals were instructed to "walk as fast as possible", which is a better predictor of peak metabolic capacity [24]. The protocol of the 6MWT consisted of 2 minutes of sitting rest, 2 minutes of standing rest, 6 minutes of walking, and 6 minutes of sitting rest in recovery, totaling 16 minutes (see Fig 1). A single physiotherapist with a background in functional tests conducted all the 6MWT.

**Oxygen uptake on- and off-kinetics.** Breath-by-breath ventilatory and metabolic variables [e.g. absolute $\dot{V}O_2$ (mL/min), relative $\dot{V}O_2$ (mL/kg/min), and respiratory exchange ratio (RER)] were measured through Oxycon Mobile$^{®}$ (Mijnhardt/Jäger, Würzburg, Germany), a valid and reliable portable metabolic analyzer [25] during the protocol of the 6MWT. The 6MWT was chosen instead of treadmill or cycle ergometer tests since this functional test accurately reflects real-world walking performance in post-stroke individuals [26] and also the metabolic response of walking on the treadmill is significantly higher both with and without support than that of walking the ground in post-stroke individuals, even at matched speeds [27, 28]. Before each test, the device was calibrated according to the manufacturer's specifications. Participants were instructed to: 1) not drink alcohol and caffeinated beverages from 24 hours prior to the test; 2) not perform any kind of physical exercise from 72 hours prior to the test; and 3) not consume a large meal from 2 hours prior to the test. Before starting the protocol of the 6MWT, participants rested sitting for 10 minutes in order to stabilize ventilatory and metabolic values.

The steady-state conditions were calculated by the standard deviation of relative $\dot{V}O_2$ over the last one minute in sitting and standing positions, and over the last three minutes of the 6MWT and recovery phase (see Fig 1). The steady-state condition was defined as the standard deviation of relative $\dot{V}O_2 \leq 2.0$ mL/kg/min and RER values $< 1.1$ [29]. Participants who did not reach the steady-state condition according to this definition were excluded from the analysis. The relative $\dot{V}O_2$ raw data were pre-processed by removing each 8-point window value above 3 standard deviations of the local mean (removing the outliers) and averaging the breath-by-breath measurements over consecutive periods of 8 breaths (moving average filter), in this order (see Fig 2) [30].

Afterwards, the on- (60 seconds of rest condition + 360 seconds of 6MWT) and off-kinetics (60 seconds of 6MWT + 360 seconds of recovery) of $\dot{V}O_2$ were analyzed by the monoexponential model following the previous literature [31]. The equations are described below. Please also see Fig 2 for details.

$$\dot{V}O_2 \text{ on-kinetics}: \quad \dot{V}O_2(t) = \dot{V}O_{2REST} + \Delta\dot{V}O_{2ON} \times (1 - e^{-(t-TD)/\tau})$$

$$\dot{V}O_2 \text{ off-kinetics}: \quad \dot{V}O_2(t) = (\Delta\dot{V}O_{2OFF} \times e^{-(t-TD)/\tau}) + \dot{V}O_{2RECOVERY}$$

Where $\dot{V}O_2(t)$ represents the $\dot{V}O_2$ at any time (t); $\dot{V}O_{2REST}$ is the resting value of $\dot{V}O_2$ in the standing position; $\dot{V}O_{2RECOVERY}$ is the recovery value of $\dot{V}O_2$ in the sitting position; $\Delta\dot{V}O_{2ON}$ is the $\dot{V}O_2$ magnitude of response at the onset of walking ($\dot{V}O_{2SS}$ - $\dot{V}O_{2REST}$); $\Delta\dot{V}O_{2OFF}$ is the $\dot{V}O_2$ magnitude of response during post-walking recovery ($\dot{V}O_{2SS}$ - $\dot{V}O_{2RECOVERY}$); TD is the time delay; and $\tau$ is the time constant of the exponential response of interest. For the $\dot{V}O_2$ on-kinetics analysis, we removed the data relative to the first 25-35s after onset (i.e. the cardiodynamic phase) [32]. Each individual curve was assessed visually by two evaluators, and therefore the time between 25 to 35 seconds with less residue was deleted. The mean response time (MRT = TD + $\tau$), i.e. the time required for $\dot{V}O_2$ to achieve 63% of the $\Delta\dot{V}O_{2ON}$ or $\Delta\dot{V}O_{2OFF}$, was corrected by the work rate (wMRT$_{ON}$ and wMRT$_{OFF}$, respectively) during the 6MWT in order to take into account the participants' individual effort and used for analysis [33]. The equations are described below:

$$wMRT_{ON} = \frac{MRT_{ON}}{VO_{2SS} - VO_{2REST}} \qquad\qquad wMRT_{OFF} = \frac{MRT_{OFF}}{\dot{VO}_{2SS} - VO_{2REST}}$$

**Physical activity level.** The physical activity level was measured by the StepWatch® Activity Monitor (SAM, Modus Health, Washington, D.C., USA), a wearable activity monitor [34, 35]. The SAM was calibrated and attached to the participants' nonparetic ankle. The participants were instructed to wear the SAM for 9 days, removing it for sleeping, swimming, and showering. The first and last days of measurements were excluded from the analyses because the device was placed and removed on these days. Participants were given an instruction sheet with detailed information about the care and use of the SAM. The mean steps/day was used to characterize the sample as a sedentary lifestyle (< 5000 steps/day), low active lifestyle (5000–7499 steps/day) and physically active lifestyle ($\geq$ 7500 steps/day) [36].

## Day 2 assessment

Body composition, arterial compliance and endothelial function were measured in sequence in the morning at visit 2, in a quiet, dimly lit and humidity and temperature-controlled room (50–60% and 22–24°C, respectively). Participants were instructed to fast overnight ($\geq$ 8h), refrain from caffeinated products ($\geq$ 12h), from vitamin supplements ($\geq$ 72h) and from moderate and vigorous physical activity ($\geq$ 48h) prior to the assessments [37–39]. All female participants were in the menopause period without hormone replacement therapy. The same physiotherapist who was experienced in day 2 assessments carried out all the exams.

**Body composition.** Weight, skeletal muscle mass (SMM) and body fat mass (BFM) were measured by a bioelectrical impedance analyzer (InBody® 720, InBody Co., Ltd., Seoul, Korea). Body mass index (BMI, kg/m²) was calculated using the following formula: BMI = [(weight in kg)/(height in m)²]. According to BMI, participants were classified as underweight

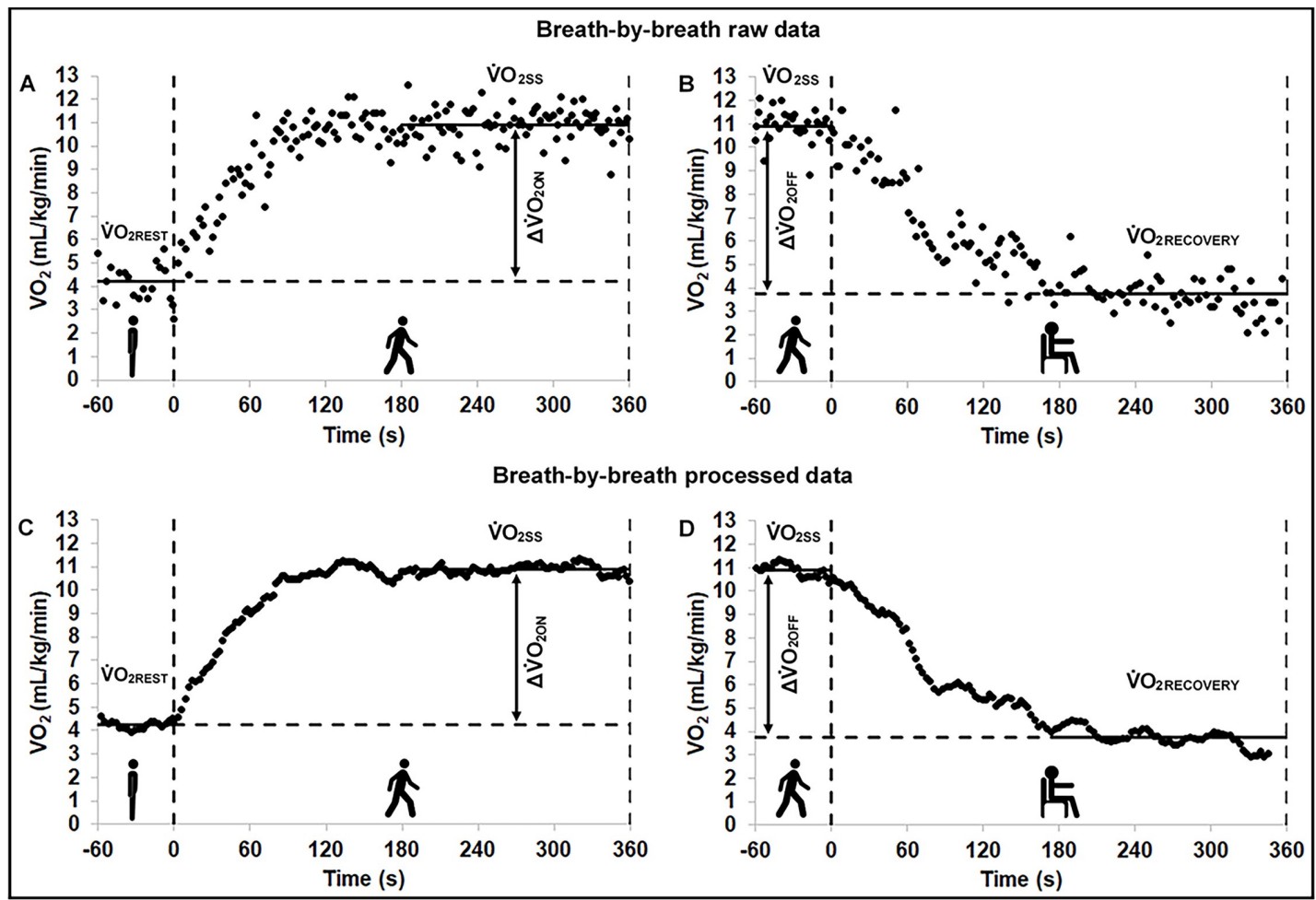

**Fig 2.** Oxygen uptake on- (A and C) and off-kinetics (B and D) response to the 6-minute walk test. Oxygen uptake raw data (A and B) measured breath-by-breath from a sixty-two-year-old woman with stroke and severe motor function impairment. The outliers (values above 3 standard deviations) were removed and a moving average filter was used by averaging the values over consecutive periods of 8 breaths (C and D). The vertical two dashed lines in each panel indicate the beginning and the end of the posture transition, in sequence. mL/kg/min, milliliter per kilogram per minute; s, second; $\dot{V}O_2$, oxygen uptake; $\dot{V}O_{2REST}$, oxygen uptake at rest; $\dot{V}O_{2RECOVERY}$, oxygen uptake during the recovery period; $\dot{V}O_{2OSS}$, oxygen uptake during effort at steady-state level; $\Delta\dot{V}O_{2ON}$, oxygen uptake on-kinetics magnitude of response ($\dot{V}O_{2SS}$ - $\dot{V}O_{2REST}$); $\Delta\dot{V}O_{2OFF}$, oxygen uptake off-kinetics magnitude of response ($\dot{V}O_{2SS}$ - $\dot{V}O_{2RECOVERY}$).

(15.0–19.9 kg/m$^2$), normal weight (20.0–24.9 kg/m$^2$), overweight (25.0–29.9 kg/m$^2$), class I obesity (30.0–34.9 kg/m$^2$), class II obesity (35.0–39.9 kg/m$^2$) and class III obesity ($\geq$ 40 kg/m$^2$) [40]. Skeletal muscle mass index (SMMI, kg/m$^2$) was calculated using the following formula: SMMI = [(skeletal muscle mass in kg)/(height in m)$^2$]. Low SMMI was defined as < 8.87 kg/m$^2$ for men and < 6.42 kg/m$^2$ for women, which are used for the diagnosis of sarcopenia [41]. Body fat mass index (BFMI, kg/m$^2$) was calculated using the following formula: BFMI = [(body fat mass in kg)/(height in m)$^2$] [42].

**Arterial compliance.** SphygmoCor® XCEL (AtCor Medical Pty. Ltd., Sydney, Australia) was used to calculate carotid-femoral pulse wave velocity (cfPWV), augmentation index (AIx), augmentation index normalized to a heart rate of 75 bpm (AIx75) and reflection magnitude (RM), measures of arterial compliance. For the measurement of cfPWV, a cuff was placed on the participant's nonparetic upper thigh. The distances from the anterior superior iliac spine to the top of the cuff, from the sternal notch to top of the cuff, and from the sternal notch to

the carotid were measured and the values were entered into the SphygmoCor software database. For the measurements of AIx, AIx75 and RM, a cuff was placed on the participant's non-paretic arm. Five successive sequences of each measurement were performed by the same evaluator in each individual in the supine position after at least 10 minutes of rest. The mean of three similar measurements with a standard deviation of less than 10% was used for analysis [37, 38]. Arterial stiffness was defined as cfPWV $\geq$ 10 m/s [37].

**Endothelial function.** Endothelial function was measured using brachial artery flow-mediated dilation (baFMD) technique, a non-invasive measure based on endothelium-dependent vasodilation [39]. The participants rested supine for 10 minutes prior to the baFMD procedure. For the measurement of baFMD, a cuff was placed on the participants' nonparetic forearm, and the arm was abducted to 90 degrees and forearm positioned in supine. An ultrasonography of the brachial artery (M-Turbo, Sonosite, Seattle, WA, USA) was used in a longitudinal plane proximal to the antecubital fossa 1–3 cm. The ultrasound probe (11 MHz) was positioned to view the anterior and posterior lumen-intimate interfaces when measuring the diameter or velocity of the central flow (pulsed Doppler). After the initial images are recorded, an anterior pressure cuff on the forearm was inflated at 220 mmHg for 5 min. To evaluate baFMD, 10 images were captured at a rate of 10 images per second for 1 min, 2 and 3 min after cuff release. Resting brachial flow velocity and peak velocity after cuff release were also recorded. The images were digitally recorded using software Brachial Analyzer (Medical Imaging Applications LLC, Coralville, Iowa, USA) and then further analyzed. The baFMD was calculated using the mean brachial artery diameter as the baseline, compared with the highest mean values obtained after forearm occlusion release using the following formula: FMD (%) = [(peak diameter–baseline diameter)/baseline diameter] x 100. Arterial dysfunction was defined as baFMD < 10% [43].

## Day 3 assessment

**Hematological and inflammatory profile.** Red blood cell (also called erythrocyte, RBC) count, and hemoglobin (Hgb) and hematocrit (Hct) concentrations were measured by an automated hematology analyzer (CELL-DYN Ruby, Abbott Laboratories, Chicago, Illinois, USA), and high-sensitivity C-reactive protein (hs-CRP) analysis was performed by using a chemistry analyzer (Abbot Architect CI 8.200, Abbott Laboratories, Chicago, Illinois, USA). A blood sample was collected from the nonparetic forearm vein in the morning after 10-12h of fasting overnight. Participants were instructed not to perform moderate and vigorous physical activity ($\geq$ 48h), not to attend the exam if any inflammatory process was present, and to maintain their usual diet prior to the exam. Individuals were also asked to report any recent symptom or event during the blood sampling week, such as ongoing or recent upper respiratory infection, recent vaccination, musculoskeletal symptoms and significant headache, and, in case of the presence of any of them, the blood collection was rescheduled. Anemia was defined as Hgb concentrations < 130 g/L for men and < 120 g/L for women [44]. The hs-CRP level was also used to characterize the sample as low- (< 1.0 mg/L), medium- (1.0–3.0 mg/L) and high-grade systemic inflammation (> 3.0 mg/L) [45].

## Data analysis

Characteristics of the sample were expressed as absolute numbers (percentage, %), means (standard deviation, SD) or medians (interquartile range, IQR). According to the Shapiro-Wilk test, the wMRT$_{ON}$ (W[24] = 0.91, p = 0.03) and wMRT$_{OFF}$ (W[24] = 0.90, p = 0.02) data showed no normality, thus nonparametric tests were used for all analyses. The Wilcoxon signed-rank test was used to determine whether there is a significant difference between baseline and recovery $\dot{V}O_2$ values and between MRT$_{ON}$ and MRT$_{OFF}$, and between wMRT$_{ON}$ and wMRT$_{OFF}$ [46].

Spearman's rank correlation coefficient ($r_s$) was used to determine whether there is a significant correlation between the wMRT$_{ON}$ and wMRT$_{OFF}$ with the following: 1) body composition (weight, BMI, BFM, BFMI, SMM and SMMI); 2) arterial compliance (cfPWV, AIx, AIx75 and RM); 3) endothelial function (baFMD); and 4) hematological (RBC, Hgb and Hct) and inflammatory (hs-CRP) profiles. The magnitude of the correlation was based on Munro's classification (low [0.26 to 0.49], moderate [0.50 to 0.69], high [0.70 to 0.89] and very high [0.90 to 1.00]) [47]. We used bias-corrected and accelerated (BCa) bootstrap resampling with 10,000 replications to estimate 95% confidence interval (CI$_{95}$). CI$_{95}$ estimates which did not include zero were considered statistically significant at the level of 5% [46].

All analyses were two-tailed and performed with a significance level of 5% using the Statistical Package for the Social Sciences, version 26.0 (SPSS Inc., Chicago, IL, USA). In addition, we used a syntax file (S1 File) to perform a non-parametric partial correlation in SPSS [48] using the variables with a significant correlation coefficient to control by confounding variables in each variables groups: 1) body composition; 2) arterial compliance; and 3) hematological and inflammatory (hs-CRP) profiles.

## Results

Four hundred and forty-three individuals were contacted to participate in the study. Out of 443 subjects, 223 were not assessed for eligibility. Thus, 220 participants were assessed for eligibility, but 176 were not included. In total, 44 participants were recruited, however twenty were excluded from the final analysis due to missing data, inability to reach the steady state during the 6MWT, and refusal to participate after initial consent. Hence, the data for 24 of these individuals were ultimately included for analysis (see Fig 3). All participants completed the 6MWT without stopping and reached the steady-state condition (please see S1-S3 Tables in S1 File), which means that they walked at a constant workload[29]. Furthermore, there were no complications during the test.

Data from twenty-four participants after chronic stroke were used for analysis. Participants were, on average, elderly (63%; $\geq$ 60 years of age), sedentary (83%; < 5000 steps/day) and overweight (54%; BMI 25.0–29.9 kg/m$^2$). Most of them had an ischemic stroke (88%) on the left side (75%) with a severe motor impairment (37%). Additionally, most of the participants did not have arterial stiffness (89%; cfPWV < 10 m/s) but had arterial dysfunction (82%; baFMD < 10%) and medium-grade systemic inflammation (59%; hs-CRP 1.0–3.0 mg/L), and none of them had anemia (100%; Hgb level $\geq$ 130 g/L for men and $\geq$ 120 g/L for women) or sarcopenia (100%; SMMI > 8.87 kg/m$^2$ for men and > 6.42 kg/m$^2$ for women) (see Table 1).

## Metabolic and $\dot{V}O_2$ kinetics response to the six-minute walk test

Most of the participants walked at a light (25%, 30–39% predicted $\dot{V}O_2$ reserve) and a moderate (50%, 40–59% predicted $\dot{V}O_2$ reserve) intensity [49, 50] during the 6MWT. Participants took almost twice as long to recover from the 6MWT (wMRT$_{OFF}$ = 0.16 min$^2$/mL/kg) than to adjust $\dot{V}O_2$ toward a steady state (wMRT$_{ON}$ = 0.10 min$^2$/mL/kg) and this difference was significant (T = 292, p < 0.001) (see Table 2).

## Relationship between the body composition and the $\dot{V}O_2$ kinetics

The relationships between the body composition and the $\dot{V}O_2$ kinetics during 6MWT are presented in Table 3. The wMRT$_{ON}$ was not correlated with any body composition variable. The wMRT$_{OFF}$ presented a low negative correlation with SMM and SMMI but did not present a correlation with any other body composition variable (please see S1 and S2 Figs in S1 File).

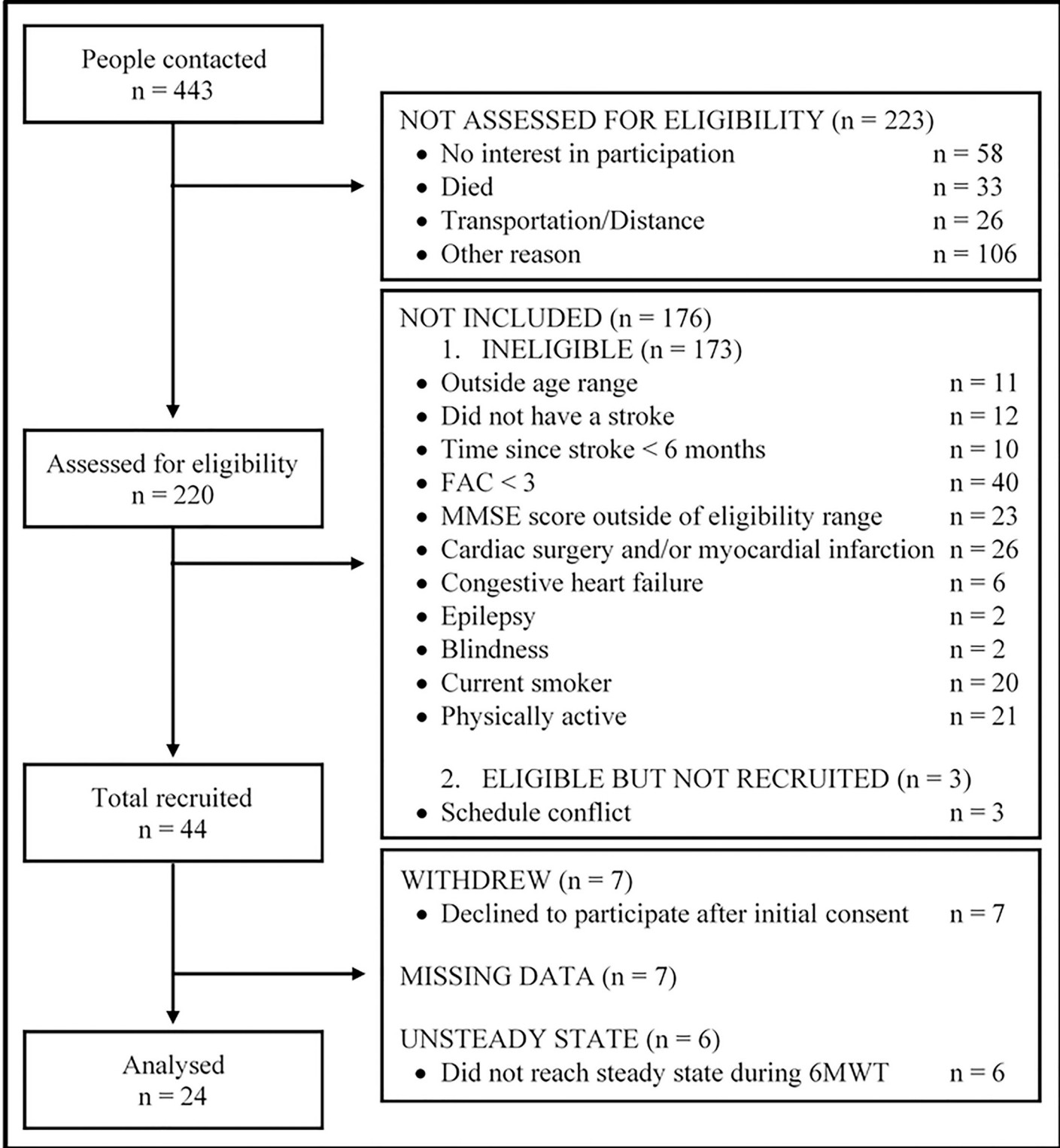

**Fig 3. Flow chart for selecting the participants for this correlational, cross-sectional pilot study.** 6MWT, six-minute walk test; FAC, functional ambulation category; MMSE, Mini-Mental State Examination.

**Table 1. Participant demographic and clinical characteristics (n = 24).**

| Characteristics | | Interval (min–max) |
|---|---|---|
| Men (n = 15) age (years), mean (SD) | 60 (11) | 44–76 |
| Women (n = 9) age (years), mean (SD) | 62 (4) | 55–68 |
| Stroke Characteristics | | |
| Time since stroke (months), median (IQR) | 41 (24 to 60) | 6–259 |
| Stroke type, n ischemic (%) | 21 (88) | |
| Lesion side, n left (%) | 18 (75) | |
| Lower Extremity Fugl Meyer Score, median (IQR) | 29 (19 to 32) | 11–34 |
| Fugl Meyer Score (Motor function), median (IQR) | 76 (33 to 97) | 11–99 |
| Slight (96–99), n (%) | 7 (29) | |
| Moderate (85–95), n (%) | 4 (17) | |
| Marked (50–84), n (%) | 4 (17) | |
| Severe (< 50), n (%) | 9 (37) | |
| 6-Minute Walk Test | | |
| Distance achieved (meters), mean (SD) | 302.63 (129.01) | 120–661.65 |
| Speed achieved (m/s), mean (SD) | 0.84 (0.36) | 0.33–1.84 |
| StepWatch[TM] Activity Monitor | | |
| Number of Steps (steps/day), median (IQR) | 3697 (2733 to 4324) | 1547–8568 |
| Body Composition | | |
| Weight (kg), mean (SD) | 75 (13) | 56–106 |
| Body Mass Index (kg/m$^2$), mean (SD) | 28.5 (4.6) | 20.9–41.4 |
| Body fat mass[a] (kg), mean (SD) | 27 (8) | 14–43 |
| Body fat mass index[a] (kg/m$^2$), mean (SD) | 10.2 (3.0) | 5.2–15.8 |
| Skeletal muscle mass[a] (kg), mean (SD) | 26 (4) | 17–35 |
| Skeletal muscle mass index[a] (kg/m$^2$), mean (SD) | 9.7 (1.0) | 7.7–11.9 |
| Arterial compliance | | |
| cfPWV[b] (m/s), median (IQR) | 7.9 (7.5 to 9.0) | 5.4–15.9 |
| AIx[a] (%), mean (SD) | 24 (11) | 2–47 |
| AIx75[a] (%), mean (SD) | 19 (11) | -4–41 |
| Reflection magnitude[a] (%), mean (SD) | 62 (10) | 43–80 |
| Endothelial function | | |
| baFMD[c] (%), mean (SD) | 5.85 (4) | -3.55–13.31 |
| Hematological and inflammatory profiles[c] | | |
| Erythrocyte (million/mm$^3$), mean (SD) | 4.95 (0.61) | 3.91–6.24 |
| Hemoglobin (g/dL), mean (SD) | 14.5 (1.4) | 11.6–16.7 |
| Hematocrit (%), mean (SD) | 43.1 (4.9) | 33.6–52.5 |
| High-sensitivity C-reactive protein (mg/L), mean (SD) | 2.24 (1.36) | 0.28–5.80 |

**Note:** Continuous variables with normal distribution are presented as means [standard deviations (SDs)]; nonnormal variables are reported as medians [interquartile ranges (IQRs)].

**Abbreviations:** %, percentage; AIx, augmentation index; AIx75, augmentation index normalized to a heart rate of 75 bpm; baFMD, brachial artery flow-mediated dilation; cfPWV, carotid-femoral pulse wave velocity; g/dL, grams per deciliter; IQR, interquartile range; kg, kilogram; kg/m$^2$, kilogram per meter$^2$; million/mm$^3$, million per cubic millimeter; SD, standard deviation; steps/min, steps per minute.

[a]n = 23.

[b]n = 19.

[c]n = 22.

**Table 2. Metabolic and $\dot{V}O_2$ kinetics response to the six-minute walk test (n = 24).**

| Variables | Sitting | Standing | Test | Recovery |
|---|---|---|---|---|
| $\dot{V}O_2$ (mL/kg/min) | 3.22 (3.00 to 3.70) | 3.52 (2.95 to 4.12) | 10.59 (8.62 to 12.15) | 3.37 (3.02 to 3.90) |
| $\Delta\dot{V}O_2$ (mL/kg/min) | 7.21 (5.69 to 8.82) | 6.84 (5.57 to 8.47) | NA | 7.03 (5.51 to 8.47) |
| Predicted $\dot{V}O_{2MAX}$ (%) | 11 (9 to 15) | 12 (9 to 17) | 38 (33 to 44) | 11 (9 to 16) |
| Predicted $\dot{V}O_2R$ (%) | NA | 14 (10 to 20) | 42 (37 to 50) | 13 (10 to 18) |
| $\dot{V}CO_2$ (mL/min) | 208.08 (196.20 to 243.41) | 247.16 (212.31 to 291.44) | 731.85 (579.18 to 841.58) | 244.46 (227.58 to 267.19) |
| RER | 0.90 (0.86 to 0.96) | 0.92 (0.85 to 1.00) | 0.95 (0.86 to 0.99) | 0.99 (0.89 to 1.05) |
| | **$\dot{V}O_2$ on-kinetics** | | **$\dot{V}O_2$ off-kinetics** | **p** |
| MRT (s) | 46 (41 to 54) | | 71 (64 to 78) | 0.01* |
| wMRT (min²/mL/kg) | 0.10 (0.09 to 0.15) | | 0.16 (0.14 to 0.21) | 0.01* |

**Note:** Variables are reported as medians (interquartile ranges). We used the Wilcoxon signed-rank test to determine whether there is a significant difference between $\dot{V}O_2$ on- and off-kinetics variables. Predicted maximal oxygen uptake [79.9 –(0.39 x age)–(13.7 x sex [0 = male; 1 = female])–(0.127 x weight [lbs])] [50]. Predicted oxygen uptake reserve [$\dot{V}O_2R = \dot{V}O_{2MAX}-\dot{V}O_2$ at rest sitting].

**Abbreviations:** %, percentage; min²/mL/kg, minute square per milliliter per kilogram; mL/kg/min, milliliter per kilogram per minute; mL/min, milliliter per minute; MRT, mean response time; NA, not applicable; RER, respiratory exchange ratio; s, second; $\dot{V}CO_2$, carbon dioxide output; $\dot{V}O_2$, oxygen uptake; $\dot{V}O_{2MAX}$, maximal oxygen uptake; $\dot{V}O_2R$, oxygen uptake reserve; wMRT, mean response time corrected for work rate; $\Delta\dot{V}O_2$, oxygen uptake magnitude of response.

*p ≤ 0.05.

There was a low negative correlation between wMRT$_{OFF}$ and SMM, when controlled by BMI (r[20] = -0.48, p = 0.03), BFM (r[20] = -0.48, p = 0.03) and BMFI (r[20] = -0.47, p = 0.03) and a correlation that approached the significance when controlled by the weight (r[20] = -0.40, p = 0.06). There was a low negative correlation between wMRT$_{OFF}$ and SMMI, when controlled by BFM (r[20] = -0.49, p = 0.02) and a correlation that approached the significance when controlled by the weight (r[20] = -0.38, p = 0.08). Moreover, there was a moderate negative correlation between wMRT$_{OFF}$ and SMMI, when controlled by BMI (r[20] = -0.53, p = 0.01) and BFMI (r[20] = -0.51, p = 0.02).

## Relationship between the arterial compliance and the $\dot{V}O_2$ kinetics

The relationships between the arterial compliance variables and the $\dot{V}O_2$ kinetics during 6MWT are presented in Table 3. The wMRT$_{ON}$ was not correlated with any of the arterial compliance variables. The wMRT$_{OFF}$ presented the following findings: 1) a low positive correlation with the percentage of AIx and the percentage of RM; and 2) a moderate positive correlation with the percentage of AIx75. There was no correlation between the wMRT$_{OFF}$ and cfPWV (please see S3 Fig in S1 File).

When controlled by cfPWV, the wMRT$_{OFF}$ presented a high positive correlation with the AIx75 (r[16] = 0.76, p < 0.01) and a correlation that approached the significance with AIx (r[16] = 0.47, p = 0.05), but there was no correlation with RM (r[16] = 0.29, p = 0.25).

## Relationship between the endothelial function and the $\dot{V}O_2$ kinetics

The relationship between the endothelial function assessed through baFMD and the $\dot{V}O_2$ kinetics during 6MWT is presented in Table 3. Neither wMRT$_{ON}$ nor wMRT$_{OFF}$ was correlated with baFMD (please see S4 Fig in S1 File).

**Table 3. Relationship between the oxygen uptake kinetics and the underlying mechanisms (n = 24).**

| Variables | wMRT$_{ON}$ (min$^2$/mL/kg) | | wMRT$_{OFF}$ (min$^2$/mL/kg) | |
|---|---|---|---|---|
| | r$_s$ [BCa CI$_{95}$] | p | r$_s$ [BCa CI$_{95}$] | p |
| Age (years) | 0.12 [-0.31, 0.50] | 0.57 | 0.20 [-0.22, 0.51] | 0.36 |
| Body Composition | | | | |
| Weight (kg) | 0.05 [-0.34, 0.42] | 0.82 | -0.23 [-0.53, 0.16] | 0.28 |
| Body mass index (kg/m$^2$) | 0.24 [-0.12, 0.55] | 0.27 | -0.01 [-0.36, 0.36] | 0.96 |
| Body fat mass[a] (kg) | 0.21 [-0.18, 0.55] | 0.33 | 0.00 [-0.37, 0.38] | 1.00 |
| Body fat mass index[a] (kg/m$^2$) | 0.25 [-0.14, 0.58] | 0.26 | 0.12 [-0.26, 0.47] | 0.59 |
| Skeletal muscle mass[a] (kg) | -0.31 [-0.59, 0.08] | 0.15 | -0.46 [-0.70, -0.10] | 0.03* |
| Skeletal muscle mass index[a] (kg/m$^2$) | -0.19 [-0.53, 0.20] | 0.37 | -0.45 [-0.71, -0.08] | 0.03* |
| Arterial compliance | | | | |
| cfPWV[b] (m/s) | 0.21 [-0.26, 0.59] | 0.38 | 0.18 [-0.33, 0.59] | 0.47 |
| AIx[a] (%) | 0.18 [-0.23, 0.55] | 0.40 | 0.44 [0.05, 0.74] | 0.04* |
| AIx75[a] (%) | 0.36 [-0.04, 0.69] | 0.09 | 0.64 [0.28, 0.87] | < 0.01* |
| Reflection magnitude[b] (%) | 0.18 [-0.27, 0.61] | 0.42 | 0.43 [0.08, 0.69] | 0.04* |
| Endothelial Function | | | | |
| baFMD[c] (%) | -0.23 [-0.59, 0.19] | 0.31 | -0.18 [-0.52, 0.22] | 0.43 |
| Hematological and inflammatory profiles[c] | | | | |
| Erythrocyte (million/mm$^3$) | -0.38 [-0.63, 0.00] | 0.08 | -0.61 [-0.76, -0.36] | < 0.01* |
| Hemoglobin (g/dL) | -0.36 [-0.65, 0.03] | 0.10 | -0.54 [-0.70, -0.23] | 0.01* |
| Hematocrit (%) | -0.27 [-0.58, 0.15] | 0.23 | -0.52 [-0.72, -0.17] | 0.01* |
| High-sensitivity C-reactive protein (mg/L) | 0.26 [-0.18, 0.60] | 0.24 | 0.58 [0.14, 0.79] | < 0.01* |
| Motor impairment (FMA) | | | | |
| Upper and lower extremities | 0.04 [-0.38, 0.43] | 0.86 | -0.29 [-0.65, 0.15] | 0.18 |
| Lower extremity | 0.04 [-0.39, 0.46] | 0.85 | -0.30 [-0.70, 0.15] | 0.16 |

**Note:** 95% bias corrected and accelerated confidence intervals reported in square brackets. Confidence intervals based on 10,000 bootstrap samples.

**Abbreviations:** %, percentage; AIx, augmentation index; AIx75, augmentation index normalized to a heart rate of 75 bpm; baFMD, brachial artery flow-mediated dilation; BCa, bias corrected accelerated; cfPWV, carotid-femoral pulse wave velocity; CI$_{95}$, 95% confidence interval; FMA, Fugl-Meyer Assessment of Motor Recovery after Stroke; g/dL, grams per deciliter; kg, kilogram; kg/m$^2$, kilogram per meter$^2$; m/s, meter per second; mg/L, milligram per liter; million/mm$^3$, million per cubic millimeter; min$^2$/mL/kg, minute square per milliliter per kilogram; r$_s$, Spearman's rank correlation coefficient; wMRT$_{OFF}$, oxygen uptake off-kinetics mean response time corrected for work rate; wMRT$_{ON}$, oxygen uptake on-kinetics mean response time corrected for work rate.

*p ≤ 0.05.

[a]n = 23.

[b]n = 19.

[c]n = 22.

### Relationship between the hematological and inflammatory profiles and the $\dot{V}O_2$ kinetics

The relationships between the hematological and inflammatory profiles and the $\dot{V}O_2$ kinetics during 6MWT are presented in Table 3. The wMRT$_{ON}$ was neither correlated with any of the hematological variables nor with hs-CRP. However, the wMRT$_{OFF}$ presented the following findings: 1) a moderate negative correlation with the number of RBC, the Hgb level, and the percentage of Hct; and 2) a moderate positive correlation with the hs-CRP level (please see S5 Fig in S1 File).

We also found a moderate negative correlation between hs-CRP and: 1) the number of RBC (r$_s$ = -0.52, BCa CI$_{95}$ [-0.82, -0.05], p = 0.01); 2) the Hgb level (r$_s$ = -0.54, BCa CI$_{95}$ [-0.78, -0.17], p = 0.01); and 3) the percentage of Hct (r$_s$ = -0.59, BCa CI$_{95}$ [-0.84, -0.20], p < 0.01)

(please see S6 Fig in S1 File). Furthermore, the wMRT$_{OFF}$ showed correlations approaching the significance with hs-CRP when controlled by: 1) the number of RBC (r[19] = 0.39, p = 0.08); 2) the Hgb level (r[19] = 0.41, p = 0.07); and 3) the percentage of Hct (r[19] = 0.40, p = 0.07). However, the wMRT$_{OFF}$ presented a low negative correlation with the number of RBC (r[19] = -0.45, p = 0.04) when controlled by hs-CRP.

### Relationship between the $\dot{V}O_2$ kinetics and the underlying mechanisms according to age

Considering a large age range, we divided the sample into two groups (adults [19–59 years] and the elderly [$\geq$ 60 years]) and carried out the aforementioned analyses in each group. Neither wMRT$_{ON}$ nor wMRT$_{OFF}$ was correlated with any variable in the adult group (please see S4 Table in S1 File). In the elderly group, the wMRT$_{OFF}$ presented a high negative correlation with SMMI and showed correlations that approached the significance with SMM, AIx75 and hs-CRP (please see S5 Table in S1 File).

## Discussion

This study investigated whether the delay in $\dot{V}O_2$ response at the onset of a short bout of walking (wMRT$_{ON}$) and during post-walking recovery (wMRT$_{OFF}$) correlated with: 1) body composition; 2) arterial compliance; 3) endothelial function; and 4) hematological and inflammatory profiles in post-stroke individuals. This study unprecedently showed that wMRT$_{OFF}$ presented correlation with SMM, SMMI, AIx, AIx75, RM, RBC, Hgb, Hct, and hs-CRP. However, the wMRT$_{ON}$ presented no correlation with the evaluated variables.

Among the mechanisms related to the $VO_2$ response kinetics, we assessed some related to the muscular and cardiovascular systems. Regarding the muscular system, our findings suggested that the skeletal muscle mass seems to play a more significant limiting role in the regulation of $\dot{V}O_2$ during the recovery phase than at the onset of walking. The highest quantity of mitochondria in our body is found in the skeletal muscle mass in order to provide substantial amounts of adenosine triphosphate (ATP), our energy currency [51]. After a stroke, the loss of skeletal muscle mass is characterized by a decrease in mitochondria-rich slow-twitch muscle fibers [52]. As the recovery process is also energy-dependent, such as the resynthesis of the intramuscular store of phosphocreatine [53], a lower number of mitochondria available means less ATP production and energy, and therefore slows down recovery.

Considering the cardiovascular system, our findings suggested that most evaluated variables (i.e. compliance and function arterial variable, and hematological and inflammatory variables) seem to play a limiting role in $\dot{V}O_2$ kinetics of poststroke individuals when walking. Nevertheless, greater distensibility of the arterial blood vessels was also related to shorter recovery time. Indeed, according to the Hagen-Poiseuille law, the blood flow rate is directly proportional to the radius to the fourth power of the vessel lumen [54], so any change in blood vessel diameter results in considerable variation in blood flow rate and, consequently, in the amount of oxygen transported. During moderate-intensity, as observed during 6MWT, $\dot{V}O_2$ kinetics may be limited by intramyocyte derangements perturbations in some chronic diseases [55]. However, for some more deconditioned patients, 6MWT could be a high-intensity exercise. In this context, other systemic disturbances, such as the reduced arterial compliance (as observed by AIx and AIx75, both in %), were related to wMRT$_{OFF}$, which could explain that the sluggish $\dot{V}O_2$ kinetic is associated to arterial stiffness present in these patients.

Furthermore, the levels of hemoglobin, hematocrit, and the erythrocytes seem to play a supporting role in the time it may take for them to recovery after walking, but a contrasting result

was found regarding the hs-CRP levels, an inflammatory biomarker. Since almost all oxygen transported from the lungs to body tissues is bound to hemoglobin [54], a higher number of red blood cells may shorten recovery time. It is noteworthy that even in non-anemic individuals, hs-CRP levels had a moderate negative correlation with the erythrocyte count. Previous studies showed that systemic inflammation may directly impair the production of erythropoietin [13, 14], a glycoprotein cytokine that stimulates erythrocyte production in the bone marrow.

In addition, the inflammatory state has been associated with insufficient production of vasodilators [56], such as nitric oxide, which impairs the endothelium-dependent vasodilation and explains the non-correlation between endothelial function (baFMD) and $\dot{V}O_2$ on- and off-kinetics response. In addition, despite the fact that previous studies showed a positive association between baFMD and peak $\dot{V}O_2$ ($\dot{V}O_{2PEAK}$) [15], it seems that once impairment of endothelial function is installed, it does not play a significant response in submaximal physical activities.

It is also worth highlighting that the age and motor function do not seem to be factors limiting the speed of the $\dot{V}O_2$ to achieve a steady state during walking or for the recovery after walking (please see Table 3 and S7 Fig in S1 File). Recently, George et al. [57] observed that aging per se does not determine the $\dot{V}O_2$ response. According to the study, even with an age-related reduction in the $\dot{V}O_{2PEAK}$, inactive elderly individuals took as long as their much younger and inactive matched counterparts to adjust to exercise. Similar to age, factors that alter the biomechanics of the body inferred from the Fugl-Meyer Assessment scale, such as spasticity and muscle co-contractions, do not relate to response $\dot{V}O_2$ kinetics. On the other hand, Ribeiro et al. [29] and Billinger et al. [58] showed correlations between the Fugl-Meyer Assessment scale and energy cost and $\dot{V}O_{2PEAK}$, respectively, other measurements of aerobic endurance [4]. These findings together reinforce the importance of a multicomponent rehabilitation program for improving physical activity tolerance in this population.

## Clinical implications of this study

Although this study has a potential mechanistic nature, our results may point to strategies that aim to accelerate these responses of $\dot{V}O_2$ kinetics, and thus reduce the deleterious effects on the bioenergetic machinery of the muscles and on the cardiovascular function of these patients. Endurance exercise training seems to be the most effective therapeutic modality for the speeding up of the $\dot{V}O_2$ kinetics response [1]. Both young people and the elderly showed a faster response to $VO_2$ kinetics after brief sessions ($\leq$ 3 sessions) of aerobic training protocols [59, 60], which earlier could decrease effort and increase tolerance during the performance of activities of daily living among the stroke individuals who meet the exercise recommendations for stroke survivors [61]. However, there is little evidence of improvement in $VO_2$ kinetics related to any type of exercise in post-stroke individuals. We found only one study [62] that observed improvement in $\dot{V}O_2$ kinetics following a low-intensity endurance training protocol. These individuals have stroke-related cardiovascular and skeletal muscle change (e.g. skeletal muscle fiber shift, and smaller peripheral artery blood flow and diameter in the stroke-affected side) [9, 15, 58] determining the $\dot{V}O_2$ kinetics response [1, 2], therefore the underlying mechanisms bearing on response $\dot{V}O_2$ kinetics during physical activity and exercise might differ from other populations.

## Study limitations

Our results must be interpreted with caution because of some limiting factors: (1) participants were chosen from a convenience sample (non-probability sampling), and therefore there is a

possibility of sample selection bias; (2) correlational study; (3) small sample size; (4) the lack of cardiac and pulmonary function assessments; and (5) the on-kinetics $\dot{V}O_2$ was measured in the standing position, and off-kinetics $\dot{V}O_2$ in the sitting position in order to ensure participant´s safety. However, this is a first exploratory study on the limiting mechanisms in bioenergetics kinetics response to walking, and we believe future research with larger and more heterogeneous samples (e.g. levels of physical activity and sedentary behavior, and types and chronicity of strokes) with different measurements [heart function (e.g. cardiac output, ejection fraction and diastolic function) and lung function (e.g. airway resistance and functional residual capacity)] is required to better understand how to improve bioenergetic kinetics response to activities of daily living. Furthermore, taking into account that the $\dot{V}O_2$ is related to gait patterns in post-stroke individuals [27, 28], it is reasonable to assess whether the gait pattern during overground walking using three-dimensional kinematics or inertial sensors is related to $\dot{V}O_2$ kinetics. It is also reasonable to consider sophisticated analyses, such as multiple regression and covariance analysis, and variables that have a direct bearing on the $\dot{V}O_2$ kinetics, such as $\dot{V}O_{2PEAK}$ [52].

## Conclusion

In conclusion, a slower $\dot{V}O_2$ off-kinetics response to walking is partially related to body composition, arterial compliance, and hematological and inflammatory profiles. Lower skeletal muscle mass, greater arterial stiffness, a reduced number of erythrocytes and higher systemic inflammation have been related to a greater amount of oxygen uptake during the recovery phase in post-stroke individuals.

## Supporting information

**S1 Database.**
(SAV)

**S2 Database.**
(XLSX)

**S1 File.**
(DOCX)

## Author Contributions

**Conceptualization:** Jean Alex Matos Ribeiro, Luciana Di Thommazo-Luporini, Aparecida Maria Catai, Audrey Borghi-Silva, Shane A. Phillips, Thiago Luiz Russo.

**Data curation:** Jean Alex Matos Ribeiro, Acson Gustavo da Silva Oliveira, Clara Italiano Monteiro, Gabriela Nagai Ocamoto.

**Formal analysis:** Jean Alex Matos Ribeiro, Acson Gustavo da Silva Oliveira, Luciana Di Thommazo-Luporini, Clara Italiano Monteiro, Gabriela Nagai Ocamoto, Aparecida Maria Catai, Audrey Borghi-Silva, Thiago Luiz Russo.

**Investigation:** Jean Alex Matos Ribeiro.

**Methodology:** Jean Alex Matos Ribeiro, Acson Gustavo da Silva Oliveira, Luciana Di Thommazo-Luporini, Clara Italiano Monteiro, Gabriela Nagai Ocamoto, Aparecida Maria Catai, Audrey Borghi-Silva, Shane A. Phillips, Thiago Luiz Russo.

**Project administration:** Jean Alex Matos Ribeiro, Acson Gustavo da Silva Oliveira, Luciana Di Thommazo-Luporini, Clara Italiano Monteiro, Gabriela Nagai Ocamoto, Aparecida Maria Catai, Audrey Borghi-Silva, Shane A. Phillips, Thiago Luiz Russo.

**Supervision:** Jean Alex Matos Ribeiro, Aparecida Maria Catai, Audrey Borghi-Silva, Thiago Luiz Russo.

**Writing – original draft:** Jean Alex Matos Ribeiro, Acson Gustavo da Silva Oliveira, Luciana Di Thommazo-Luporini, Clara Italiano Monteiro, Gabriela Nagai Ocamoto, Aparecida Maria Catai, Audrey Borghi-Silva, Shane A. Phillips, Thiago Luiz Russo.

**Writing – review & editing:** Jean Alex Matos Ribeiro, Acson Gustavo da Silva Oliveira, Luciana Di Thommazo-Luporini, Clara Italiano Monteiro, Gabriela Nagai Ocamoto, Aparecida Maria Catai, Audrey Borghi-Silva, Shane A. Phillips, Thiago Luiz Russo.

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
