## [Decision Letter · Decision Letter 0]

27 Aug 2020

PONE-D-20-21333

Underlying mechanisms of oxygen uptake kinetics in chronic post-stroke individuals: a correlational, cross-sectional pilot study

PLOS ONE

Dear Dr. Ribeiro,

Thank you for submitting your manuscript to PLOS ONE. After careful consideration, we feel that it has merit but does not fully meet PLOS ONE’s publication criteria as it currently stands. Therefore, we invite you to submit a revised version of the manuscript that addresses the points raised during the review process.

We look forward to receiving your revised manuscript.

Kind regards,

Yuji Ogura, Ph.D.

Academic Editor

PLOS ONE

Journal Requirements:

Additional Editor Comments (if provided):

Both experts raised critical concerns especially regarding the weakness of study design and data analysis. They thought that the conclusion of this manuscript is not supported by the results at the present form, whereas they found that this manuscripts has some merit. Please address all the concerns of the two reviewers and submit a revised version of the manuscript.

Reviewers' comments:

Reviewer's Responses to Questions

**Comments to the Author**

1. Is the manuscript technically sound, and do the data support the conclusions?

Reviewer #1: Partly

Reviewer #2: Partly

2. Has the statistical analysis been performed appropriately and rigorously? 

Reviewer #1: N/A

Reviewer #2: Yes

3. Have the authors made all data underlying the findings in their manuscript fully available?

Reviewer #1: No

Reviewer #2: Yes

4. Is the manuscript presented in an intelligible fashion and written in standard English?

Reviewer #1: Yes

Reviewer #2: No

5. Review Comments to the Author

Reviewer #1: The paper is focused on the interesting research issue where the oxygen consumption for walking in post-stroke patients. The paper is generally well written, however, there are several concerns with the current manuscript. In particular, the research design may include potential limitation.

Major comments

1. Author should clearly document the validity of the study design. As numerous papers demonstrate that the energy cost of "6-min walk test" was depending on the gait pattern such as walking speed, cadence, stride length, and were depended on the physical characters including disease condition. Thus, the authors should document the reason for using the "6-min walk test" rather than the controlled exercise (such as treadmill walking, or cycle ergometer)

2. As mentioned above, the VO2 for "6-min walk test" was associated with health status. Therefore, the authors should firstly demonstrate the gat pattern of the "6-min walk test" was independent from the obtained variables (SMI, PWV, inflammatory, etc..)

3. The conclusion of the present investigation was not supported by the finding. The relationship of variables with physical activity levels was not demonstrated.

Minor

1. The validity/accuracy of the gas analyzer should be documented.

2. The speed and gait pattern for "6-min walk test" should be shown.

3. The correlations should be demonstrated by illustrations NOT only P value.

4. Authors should be use the partial correlation, as the variables are not independent.

5. As the present study is cross-sectional analysis, the conclusion should be limited to "relationship" or "correlation" NOT "explain".

Reviewer #2: PLOS One review

The authors present the results of a correlative pilot study on onset and offset kinetics in individuals who survived a stroke event.

Comments:

1. Introduction needs a description of a mechanisms on how the stroke event actually causes these changes that affect the O2 kinetics. Is the kinetics worse because of poor health that contributed to stroke and age or did the stroke event induce these deleterious changes that were not present before the event? For examples how does a stroke event contribute to muscle fiber shift, muscle atrophy endothelial dysfunction etc.?

2. Methods

- “This is a correlational, cross-sectional pilot study with a convenience sample.” This sentence is unclear.

- Considering a large age range, could the analyses be done separately for the older (>65) and younger individuals (<45) in order to check the age effect? If the differences are insignificant, the results can be placed in the supplementary material.

3. Limitations

- Correlative study is also a limitation.

4. Conclusions

- Correlations cannot be used to explain the biological reactions. The manuscript needs the wording adjustment to state these conclusions more cautiously.

5. Figures

- The consort diagram should be presented first as it was described first too.

6. PLOS authors have the option to publish the peer review history of their article (what does this mean?). If published, this will include your full peer review and any attached files.

Reviewer #1: No

Reviewer #2: No

---

## [Author Response · Author response to Decision Letter 0]

28 Sep 2020

Reviewers’ Comments to the Authors:

Reviewer #1: The paper is focused on the interesting research issue where the oxygen consumption for walking in post-stroke patients. The paper is generally well written, however, there are several concerns with the current manuscript. In particular, the research design may include potential limitation.

Major comments

1. Author should clearly document the validity of the study design. As numerous papers demonstrate that the energy cost of "6-min walk test" was depending on the gait pattern such as walking speed, cadence, stride length, and were depended on the physical characters including disease condition. Thus, the authors should document the reason for using the "6-min walk test" rather than the controlled exercise (such as treadmill walking, or cycle ergometer).

Authors’ Response: As suggested by the reviewer, the reason for using the 6MWT rather than the controlled exercise is described in the methods section. Although several studies demonstrate that treadmill walking increases biomechanical symmetry in post-stroke individuals, it is accompanied by higher metabolic demands both with and without support, even at matched speeds (Brouwer et al., 2009; IJmker et al., 2013). Due to unease or instability associated with treadmill walking, post-stroke individuals present higher cadence, shorter stance time and shorter strides using it when compared to overground. Moreover, even though handrails are used, there is a greater contribution of the muscle activation pattern associated with the paretic limb during treadmill walking compared to walking overground. All in all, it is reasonable to hypothesize that the six-minute test, a functional test, reflects better the metabolic response during activities of daily living related to walking rather than treadmill walking. 

Page 6: “The 6MWT was chosen instead of treadmill or cycle ergometer tests since this functional test accurately reflects real-world walking performance in post-stroke individuals26 and also the metabolic response of walking on the treadmill is significantly higher both with and without support than that of walking the ground in post-stroke individuals, even at matched speeds27,28.”

References

26 Fulk GD, He Y, Boyne P, Dunning K. Predicting home and community walking activity poststroke. Stroke. 2017;48(2):406-411. doi: 10.1161/STROKEAHA.116.015309.

27 Brouwer B, Parvataneni K, Olney SJ. A comparison of gait biomechanics and metabolic requirements of overground and treadmill walking in people with stroke. Clin Biomech. 2009;24(9):729-734. doi: 10.1016/j.clinbiomech.2009.07.004.

28 Ijmker T, Houdijk H, Lamoth CJ, Jarbandhan AV, Rijntjes D, Beek PJ, et al. Effect of balance support on the energy cost of walking after stroke. Arch Phys Med Rehabil. 2013;94(11):2255-2261. doi: 10.1016/j.apmr.2013.04.022.

2. As mentioned above, the VO2 for "6-min walk test" was associated with health status. Therefore, the authors should firstly demonstrate the gait pattern of the "6-min walk test" was independent from the obtained variables (SMI, PWV, inflammatory, etc..).

Authors’ Response: As suggested by the reviewer, we analyzed the correlations between motor function (data about gait patterns were not collected) and the obtained variables (please see Table below). Except for body fat mass and body fat mass index (which did not correlate with VO2 kinetics), the obtained variables have not been correlated with motor function. However, we agree that this aspect is relevant and have included it as a limitation of the study. The analysis of gait patterns using 3D kinematics or inertial sensors during the test could help advance this topic, as it is clinically relevant. 

Page 20: “Furthermore, taking into account that the V̇O2 is related to gait patterns in post-stroke individuals27,28, it is reasonable to assess whether the gait pattern during overground walking using three-dimensional kinematics or inertial sensors is related to V̇O2 kinetics.”

3. The conclusion of the present investigation was not supported by the finding. The relationship of variables with physical activity levels was not demonstrated.

Authors’ Response: We agree with the reviewer’s assessment. We removed the statement “which might imply in intolerance to physical activity” from the abstract and conclusion section of the main text.

Minor

1. The validity/accuracy of the gas analyzer should be documented.

Authors’ Response: Thank you for your suggestion. The validity/accuracy of the gas analyzer has now been documented. 

Page 6: “Breath-by-breath ventilatory and metabolic variables [e.g. absolute V̇O2 (mL/min), relative V̇O2 (mL/kg/min), and respiratory exchange ratio (RER)] were measured using an Oxycon Mobile® (Mijnhardt/Jäger, Würzburg, Germany), a valid and reliable portable metabolic analyzer25 during the protocol of the 6MWT.”

Reference

25 Rosdahl H, Gullstrand L, Salier-Eriksson J, Johansson P, Schantz P. Evaluation of the oxycon mobile metabolic system against the Douglas bag method. Eur J Appl Physiol. 2010;109(2):159-171. doi: 10.1007/s00421-009-1326-9.

2. The speed and gait pattern for "6-min walk test" should be shown.

Authors’ Response: Thank you for this suggestion. The data about speed during the 6MWT have been added to Table 1. However, data about gait patterns were not collected, instead we collected data about the motor function after stroke by the Fugl-Meyer Assessment of Motor Recovery after Stroke. 

3. The correlations should be demonstrated by illustrations NOT only P value.

Authors’ Response: We agree with the suggestion. We have also represented all correlations graphically in the Supplementary Material (Figures S1 to S7).

4. Authors should be use the partial correlation, as the variables are not independent.

Authors’ Response: Thank you for pointing this out. According to the Shapiro-Wilk test, the wMRTON (W[24] = 0.908, p = 0.032) and wMRTOFF (W[24] = 0.896, p = 0.017) data showed no normality, therefore we used a syntax file to perform a non-parametric partial correlation in SPSS (https://www.ibm.com/support/pages/partial-rank-correlations-spss).

Syntax file

NONPAR CORR

/MISSING = LISTWISE

/MATRIX OUT(*).

RECODE rowtype_ ('RHO'='CORR') .

PARTIAL CORR

/significance = twotail

/MISSING = LISTWISE

/MATRIX IN(*).

Page 17: “According to the Shapiro-Wilk test, the wMRTON (W[24] = 0.91, p = 0.03) and wMRTOFF (W[24] = 0.90, p = 0.02) data showed no normality, thus nonparametric tests were used for all analyses”

Page 17: “In addition, we used a syntax file (Supplementary Material) to perform a non-parametric partial correlation in SPSS48 using the variables with a significant correlation coefficient to control by confounding variables in each variable groups: 1) body composition; 2) arterial compliance; and 3) hematological and inflammatory (hs-CRP) profiles.”

Page 14: “There was a low negative correlation between wMRTOFF and SMM, when controlled by BMI (r[20] = -0.48, p = 0.03), BFM (r[20] = -0.48, p = 0.03) and BMFI (r[20] = -0.47, p = 0.03), and a correlation that approached the significance when controlled by the weight (r[20] = -0.40, p = 0.06). There was a low negative correlation between wMRTOFF and SMMI, when controlled by BFM (r[20] = -0.49, p = 0.02) and a correlation that approached the significance when controlled by the weight (r[20] = -0.38, p = 0.08). Moreover, there was a moderate negative correlation between wMRTOFF and SMMI, whilst controlling for BMI (r[20] = -0.53, p = 0.01) and BFMI (r[20] = -0.51, p = 0.02).

Page 15: “When controlled by cfPWV, the wMRTOFF presented a high positive correlation with the AIx75 (r[(16] = 0.76, p < 0.01) and and a correlation that approached the significance AIx (r[16] = 0.47, p = 0.05), but there was no correlation with RM (r[16] = 0.29, p = 0.25).”

Page 15: “Furthermore, the wMRTOFF showed correlations that approached the significance with hs-CRP when controlled by: 1) the number of RBC (r[19] = 0.39, p = 0.08); 2) the Hgb level (r[19] = 0.41, p = 0.07); and 3) the percentage of Hct (r[19] = 0.40, p = 0.07). However, the wMRTOFF presented a low negative correlation with the number of RBC (r[(19] = -0.45, p = 0.04) when controlled by hs-CRP.”

Page 20: “It is also reasonable to consider sophisticated analyses, such as multiple regression and covariance analysis, and variables that have a direct bearing on the V̇O2 kinetics, such as the V̇O2PEAK52.”

Reference

48 International Business Machines Corporation (IBM). Partial rank correlations in SPSS. Available from: https://www.ibm.com/support/pages/partial-rank-correlations-spss. [Accessed Sep 25th, 2020].

5. As the present study is cross-sectional analysis, the conclusion should be limited to "relationship" or "correlation" NOT "explain".

Authors’ Response: To further balance the implications of our results with the potential limitations of a cross-sectional study, we changed “explained” to “related” in the first sentence of the conclusion in the abstract and conclusion section of the main text.

Page 1: “A greater amount of oxygen uptake during post-walking recovery is partially related to lower skeletal muscle mass, greater arterial stiffness, reduced number of erythrocytes and higher systemic inflammation in post-stroke individuals.”

Page 20: “In conclusion, the slower V̇O2 off-kinetics response to walking is partially related to body composition, arterial compliance, and hematological and inflammatory profiles.”

Reviewer #2: The authors present the results of a correlative pilot study on onset and offset kinetics in individuals who survived a stroke event.

Comments:

1. Introduction needs a description of a mechanisms on how the stroke event actually causes these changes that affect the O2 kinetics. Is the kinetics worse because of poor health that contributed to stroke and age or did the stroke event induce these deleterious changes that were not present before the event? For examples how does a stroke event contribute to muscle fiber shift, muscle atrophy endothelial dysfunction etc.?

Authors’ Response: The references used in the Introduction section were about stroke-related muscular and cardiovascular changes and not about poor health-related changes following stroke, but indeed it was unclear. Therefore, we have made some changes in order to clarify this. 

Page 3: “After a stroke, these individuals have deleterious stroke-related skeletal muscle changes, such as a shift from type I to type II fibers, muscle atrophy, intramuscular fat, and muscle fibrosis.8-11 In addition, they have stroke-related cardiovascular changes, such as endothelial dysfunction, impaired arterial compliance, and increased proinflammatory markers,…” 

References

8 Sions JM, Tyrell CM, Knarr BA, Jancosko A, Binder-Macleod SA. Age- and stroke-related skeletal muscle changes a review for the geriatric clinician. J Geriatr Phys Ther. 2012;35(3):155–161. doi: 10.1519/JPT.0b013e318236db92.

9 Billinger SA, Coughenour E, MacKay-Lyons MJ, Ivey FM. Reduced cardiorespiratory fitness after stroke: biological consequences and exercise-induced adaptations. Stroke Res Treat. 2012;2012:959120. doi: 10.1155/2012/959120.

10 Silva-Couto MA, Prado-Medeiros CL, Oliveira AB, Alcântara CC, Guimarães AT, Salvini TF, et al. Muscle atrophy, voluntary activation disturbances, and low serum concentrations of IGF-1 and IGFBP-3 are associated with weakness in people with chronic stroke. Phys Ther. 2014;94(7):957–967. doi: 10.2522/ptj.20130322.

11 Faturi FM, Santos GL, Ocamoto GN, Russo TL. Structural muscular adaptations in upper limb after stroke: a systematic review. Top Stroke Rehabil. 2019;26(1):73–79. doi: 10.1080/10749357.2018.1517511.

12 Eikelboom JW, Hankey GJ, Baker RI, McQuillan A, Thom J, Staton J, et al. C-reactive protein in ischemic stroke and its etiologic subtypes. J Stroke Cerebrovasc Dis. 2003;12(2):74–81. doi: 10.1053/jscd.2003.16.

2. Methods

- “This is a correlational, cross-sectional pilot study with a convenience sample.” This sentence is unclear.

Authors’ Response: We have added the information “there was no random selection” in order to make the sentence clearer.

Page 4: “This is a correlational, cross-sectional pilot study with a convenience sample (there was no random selection).”

- Considering a large age range, could the analyses be done separately for the older (>65) and younger individuals (<45) in order to check the age effect? If the differences are insignificant, the results can be placed in the supplementary material.

Authors’ Response: We appreciate this comment and agree that the analyses could be done separately in order to check the age effect. It was not possible to divide the sample into older (> 65) and younger (< 45) as we had only one individual aged below 45 years old. However, we were able to divide the sample into adults (19-59 years) and the elderly (≥ 60 year). Thus, we performed the analyses in each group (see tables below) and added the following information to the main text.

Page 16: “Considering a large age range, we divided the sample into two groups (adults [19-59 years] and the elderly [≥ 60 years]) and carried out the aforementioned analyses in each group. Neither wMRTON nor wMRTOFF was correlated with any variable in the adult group (please see Supplementary Material, Table S4). In the elderly group, the wMRTOFF presented a high negative correlation with SMMI and showed correlations that approached the significance with SMM, AIx75 and hs-CRP (please see Supplementary Material, Table S5).”

3. Limitations

- Correlative study is also a limitation.

Authors’ Response: We agree that this is a potential limitation of the study. We have added the suggested content to the manuscript on the Study limitations section.

Page 19: “Our results must be interpreted with caution because of some limiting factors: (1) participants were chosen from a convenience sample (non-probability sampling), and therefore there is a possibility of a sample selection bias; (2) correlational study;…”

4. Conclusions

- Correlations cannot be used to explain the biological reactions. The manuscript needs the wording adjustment to state these conclusions more cautiously.

Authors’ Response: Thank you for pointing this out. As suggested by the reviewer, we have made changes throughout the manuscript to make adjustments to state these conclusions more cautiously. The main modifications were the following:

1. We have removed these statements from the Discussion section

“It explained 21% of the variation in the time it may take for them to recovery after 6MWT.”

“About 18-41% of the variation in recovery time after walking was explained by the degree of arterial compliance.”

2. We have replaced the statement “Furthermore, the levels of hemoglobin, hematocrit, and the erythrocytes count explained 27-37% of the variation in the time it may take for them to recovery after walking, and a similar result (34%) was found regarding the hs-CRP levels, an inflammatory biomarker.” by

Page 17: “Furthermore, the levels of hemoglobin, hematocrit, and the erythrocytes seem to play a supporting role in the time it may take for them to recovery after walking, but a contrasting result was found regarding the hs-CRP levels, an inflammatory biomarker”

3. We have replaced the statement “It is noteworthy that even in non-anemic individuals, hs-CRP levels explained 27% of the variation in the erythrocyte count.” by

Page 17: “It is noteworthy that even in non-anemic individuals, hs-CRP levels had a moderate negative correlation with the erythrocyte count.”

5. Figures

- The consort diagram should be presented first as it was described first too.

Authors’ Response: We appreciate the reviewer’s feedback. We have made certain that the Figures are presented in the order they have been described. 

Pages 2, 6 and 7: Figure 1 (Oxygen uptake response to the 6-minute test.)

Page 7: Figure 2 [Oxygen uptake on- (A and C) and off-kinetics (B and D) response to the 6-minute walk test.]

Page 13: Figure 3 (Flow chart for selecting the participants for this correlational, cross-sectional pilot study.)

---

## [Decision Letter · Decision Letter 1]

22 Oct 2020

Underlying mechanisms of oxygen uptake kinetics in chronic post-stroke individuals: a correlational, cross-sectional pilot study

PONE-D-20-21333R1

Dear Dr. Ribeiro

We’re pleased to inform you that your manuscript has been judged scientifically suitable for publication and will be formally accepted for publication once it meets all outstanding technical requirements.

Kind regards,

Yuji Ogura, Ph.D.

Academic Editor

PLOS ONE

Additional Editor Comments (optional):

Reviewers' comments:

Reviewer's Responses to Questions

**Comments to the Author**

1. If the authors have adequately addressed your comments raised in a previous round of review and you feel that this manuscript is now acceptable for publication, you may indicate that here to bypass the “Comments to the Author” section, enter your conflict of interest statement in the “Confidential to Editor” section, and submit your "Accept" recommendation.

Reviewer #1: All comments have been addressed

Reviewer #2: All comments have been addressed

2. Is the manuscript technically sound, and do the data support the conclusions?

Reviewer #1: Yes

Reviewer #2: (No Response)

3. Has the statistical analysis been performed appropriately and rigorously? 

Reviewer #1: Yes

Reviewer #2: Yes

4. Have the authors made all data underlying the findings in their manuscript fully available?

Reviewer #1: Yes

Reviewer #2: (No Response)

5. Is the manuscript presented in an intelligible fashion and written in standard English?

Reviewer #1: Yes

Reviewer #2: Yes

6. Review Comments to the Author

Reviewer #1: The current manuscript is appropriately improved according to the reviewer comments.

I have no additional comments.

Reviewer #2: (No Response)

7. PLOS authors have the option to publish the peer review history of their article (what does this mean?). If published, this will include your full peer review and any attached files.

Reviewer #1: No

Reviewer #2: No

---

## [Editor Report · Acceptance letter]

27 Oct 2020

PONE-D-20-21333R1 

Underlying mechanisms of oxygen uptake kinetics in chronic post-stroke individuals: a correlational, cross-sectional pilot study 

Dear Dr. Ribeiro:

I'm pleased to inform you that your manuscript has been deemed suitable for publication in PLOS ONE. Congratulations! Your manuscript is now with our production department. 

Kind regards, 

on behalf of

Dr. Yuji Ogura 

Academic Editor

PLOS ONE